# Cyclodextrin’s Effect on Permeability and Partition of Nortriptyline Hydrochloride

**DOI:** 10.3390/ph16071022

**Published:** 2023-07-19

**Authors:** Tatyana Volkova, Olga Simonova, German Perlovich

**Affiliations:** G.A. Krestov Institute of Solution Chemistry RAS, 153045 Ivanovo, Russia; ors@isc-ras.ru (O.S.); glp@isc-ras.ru (G.P.)

**Keywords:** nortriptyline hydrochloride, HP-β-CD, SBE-β-CD, membrane permeability, PermeaPad barrier, distribution

## Abstract

Cyclodextrin-based delivery systems have been intensively used to improve the bioavailability of drugs through the modification of their pharmaceutically relevant properties, such as solubility, distribution and membrane permeation. The present work aimed to disclose the influence of HP-β-CD and SBE-β-CD on the distribution and permeability of nortriptyline hydrochloride (NTT•HCl), a tricyclic antidepressant drug. To this end, the distribution coefficients in the 1-octanol/buffer and n-hexane/buffer model systems and the coefficients of permeability through the cellulose membrane and lipophilic PermeaPad barrier were determined at several cyclodextrin concentrations. The results demonstrated a dramatic decrease in both the distribution and the permeability coefficients as the cyclodextrin concentration rose, with the decrease being more pronounced in SBE-β-CD due to the charge–charge attraction and electrostatic interactions between NTT and SBE-β-CD. It is these interactions that were shown to be responsible for the greater value of the constant of NTT’s association with SBE-β-CD than that with HP-β-CD. The findings of this study revealed similar trends in the 1-octanol/buffer 6.8 pH distribution and permeability through the PermeaPad barrier in the presence of CDs. These results were attributed to the determinative role of the distribution coefficient (serving as a descriptor) in permeation through the PermeaPad barrier modeling the lipophilic nature of biological barriers.

## 1. Introduction

The design of advantageous systems for the delivery of drugs to the site of action is one of the most rapidly developing areas of pharmaceutical science and technology [1]. Such systems serve to provide a sufficient rate and concentration of a drug in order to achieve the maximum therapeutic effect and minimum side effects. These aims can be achieved through the optimization of pharmacologically relevant transport properties of drug compounds, such as solubility, lipophilicity and membrane permeability, using pharmaceutical excipients. The role of pharmaceutical excipients is often played by, among other compounds, cyclodextrins–cyclic oligosaccharides possessing a hydrophilic exterior surface (facilitating solubility growth) and a nonpolar hydrophobic interior cavity (capable of the inclusion of lipophilic substances) [2]. Cyclodextrins are primarily used to enhance aqueous solubility, physicochemical stability and bioavailability through the formation of inclusion complexes with drugs. Their other applications include preventing drug–drug interactions, converting liquid drugs into microcrystalline powders, minimizing gastrointestinal and ocular irritation and reducing or eliminating unpleasant tastes and smells [3]. Cyclodextrins have also been reported to affect the stabilization of unilamellar vesicles for a tunable drug delivery depot [4,5]. However, being an advantageous tool in solubility enhancement, cyclodextrins often act as permeability reducing agents [6]. Notably, the medium pH strongly affects the drug–cyclodextrin interaction in the case of ionizable drugs as a result of the stronger affinity of neutral lipophilic molecules to the hydrophobic cyclodextrin cavity as compared to charged species [7]. The permeation of the ionized species also differs from that of the uncharged ones, because the charged solute has a solvation shell composed of water molecules when entering the hydrophobic membrane [8].

The permeability of drugs in vitro can be evaluated with the help of various types of artificial membranes. Among them, membranes composed of regenerated cellulose with different MWCO are often used to reveal the effect of auxiliary agents (for example, cyclodextrins) on the diffusion process [2]. Being applicable for diffusion rate evaluations, these barriers are hydrophilic and cannot simulate the lipophilic layer of cell membranes. The latter is achieved by utilizing phospholipid-based membranes, such as the innovative PermeaPad barrier developed by di Cagno and Bauer-Brandl [9]. As was reported, the PermeaPad barrier can be applied to estimate the permeability of drugs in the presence of cyclodextrins [10].

The distribution coefficient in the 1-octanol/water system (lipophilicity) serves as a descriptor that determines the drug’s ability to penetrate the intestinal barriers. In its turn, the partition between n-hexane (as the model of non-polar tissues) and a water medium characterizes blood/brain permeability [11]. Membrane permeability and distribution are often intercorrelated [12]. Their correlations are mostly observed when cyclodextrins are present in the aqueous phase of the distribution or permeation system [13,14]. However, this is not always the case. In our previous study [15], the permeability of the model compound iproniazid was enhanced with methylated cyclodextrin, whereas the distribution coefficient was reduced.

The object of the present study—NTT•HCl (Figure 1)—is an antidepressant drug that must be delivered to the brain in a timely manner (i.e., it must penetrate the intestinal membranes and the blood–brain barrier). To obtain deeper insight into the compound’s distribution and permeation, we state the following aims of the present study: (1) to investigate the effect of pH and cyclodextrins on the distribution of NTT•HCl in the 1-octanol/buffer and n-hexane/buffer systems; (2) to evaluate the influence of cyclodextrins on the permeation rate of the compound; and (3) to reveal the effect of the lipophilic layer in the membrane on the permeability of the compound with and without cyclodextrins.

To achieve these goals, we carried out distribution experiments at three concentrations of 2-hydroxypropyl-β-cyclodextrin (HP-β-CD) and sulfobutylether-β-cyclodextrin (SBE-β-CD) and two pH values of the buffer phase of the distribution systems (pH of 4.0 and pH of 6.8). Based on the 1-octanol/buffer distribution coefficients at different CD concentrations, we determined the association constants of the NTT•HCl/HP-β-CD and NTT•HCl/SBE-β-CD complexes via the phase distribution method [16,17]. The permeability coefficients were measured through the artificial regenerated cellulose (MWCO 12–14 kDa (RC)) membrane and the biomimetic PermeaPad barrier (PP) composed of a phosphatidylcholine layer immobilized between two cellulose supports.

Therefore, the present study is a logical continuation of our works devoted to disclosing the influence of the nature and specific features of the drug and the pharmaceutical excipient, as well as to disclosing the membrane’s properties on the distribution and permeability processes. We hope the presented results improve our understanding of drug delivery using cyclodextrin-containing systems.

The structures of the investigated compound and cyclodextrins are presented in Figure 1.

## 2. Results and Discussion

### 2.1. Effect of HP-β-CD and SBE-β-CD on the Distribution of NTT•HCl

The distribution coefficients of NTT•HCl were determined in the 1-octanol/buffer (with *C*_CD_ = 0, 0.0115, 0.025 and 0.035 M) and n-hexane/buffer (with *C*_CD_ = 0, and 0.0115 M) systems at a pH of 4.0 and a pH of 6.8 of the buffer phase. The choice of these buffers for the experiments was governed by two reasons. On the one hand, these pH values are characteristic of intestinal fluids, in which most of the drug absorption takes place. On the other hand, different ionization degrees of NTT in these media allowed us to reveal the effect of the ionization state on the investigated processes. The results are listed in Table 1. The experimental molar concentrations of NTT•HCl in the organic and aqueous phases are given in Appendix A.

First of all, the distribution coefficients of NTT•HCl were shown to be noticeably lower than those of the structural analog—tricyclic antidepressant drug amitryptiline hydrochloride (AMT•HCl) [14]: 13-fold and 7-fold in the 1-octanol/buffer pH of 6.8 and 1-octanol/buffer pH of 4.0, respectively. Bearing in mind the positive induction effect of the electron donor CH_3_ group acting as a hydrophobic substituent (increasing the lipophilicity), the lower value of the NTT•HCl distribution coefficient can be attributed to the presence of only one methyl group in the structure of the NTT amide substituent instead of two methyl groups present in the AMT structure.

The diagram in Figure 2 illustrates the influence of the 0.0115 M CD (HP-β-CD and SBE-β-CD) concentration in the aqueous phase on the NTT•HCl transition between the organic phase (1-octanol and n-hexane) and buffer (pH of 4.0 and pH of 6.8). In addition, the figure also represents the Δlog*D* parameter serving as a measure of hydrogen bonding contribution to the transport process [6]. As follows from Table 1 and Figure 2, the distribution coefficients of NTT•HCl are higher in the aqueous phase with a buffer pH of 6.8 than those in the aqueous phase with a buffer pH of 4.0 in both investigated systems: 1.33-fold and 1.61-fold for 1-octanol and n-hexane, respectively. This results from the small number of uncharged species in the buffer pH of 6.8, according to *pK*_a_ = 9.23 [18] (Appendix A). As Figure 2 shows, at both pH values the buffer → organic solvent transfer processes slow down when cyclodextrins are added to the aqueous phase. Moreover, for the 1-octanol/buffer 4.0 pH system, the distribution equilibrium is shifted to the aqueous phase in the presence of both CDs. The effect of SBE-β-CD is more pronounced as a result of the charge–charge attraction and electrostatic interactions between the charges of the cationic species of NTT and SBE-β-cyclodextrin [19]. Notably, this effect is less pronounced in the partition systems with a buffer pH of 6.8 with a small number of uncharged NTT species.

For the sake of clarity, the impact of pH on the distribution without and with a 0.0115 M concentration of CDs is illustrated in Figure 3 as the ratios between the distribution coefficients at two pH values in a particular system.

Figure 3 clearly demonstrates a dramatic increase in the NTT•HCl distribution to the 1-octanol phase in the system with a buffer pH of 6.8 as compared to that with a buffer pH of 4.0 with both CDs of the 0.0115 M concentration, whereas this effect is not pronounced for the n-hexane/buffer systems. This means that the distribution of NTT•HCl (in drug formulations containing cyclodextrins) between the hydrophilic and lipophilic media of biological objects is expected to vary in the segments of the gastro-intestinal tract with different pH values. The Δlog*D* parameter values diminishing in the presence of cyclodextrins (Table 1, Figure 2) allow us to suggest that it is the weaker effect of the specific interaction (hydrogen bonding) on the transfer process in the cyclodextrin-containing systems that, among other factors, leads to the distribution coefficient’s reduction. In addition, this effect is again more pronounced in the system with a buffer pH of 4.0, in which all particles are ionized. Notably, in all the n-hexane/buffer systems, the equilibrium is shifted to the aqueous phases, indicating that the investigated compound can have preferences for hydrophilic biological fluids rather than non-polar tissues (e.g., the brain).

Figure 4 illustrates variations in the distribution coefficients in the 1-octanol/buffer systems with the CD concentration growth. The diagram (Figure 4) shows that the maximal reduction in the distribution coefficients is observed at the maximal CD concentrations with both pH values and cyclodextrins in the aqueous phase. In addition, the minimal $D_{app}^{oct/buf}$ value is characteristic of the 1-octanol/buffer (pH of 4.0 + 0.035 M SBE-β-CD) system.

To quantitatively evaluate the influence of cyclodextrins on the transfer processes, we applied the distribution coefficients in the 1-octanol/buffer (pH of 6.8 and pH of 4.0) systems at different CD concentrations to calculate the association constants (Equations (4) and (5)) of NTT•HCl with cyclodextrins via the phase distribution method, as fully described in the literature [17,20,21]. The plots of the dependencies according to Equation (5) are given in Figure 5.

The association constants (*K*_C_) and the regression parameters of the linear dependencies illustrated in Figure 5 are listed in Table 2. For better visualization of the effect of pH and cyclodextrin’s nature on the stability of the complexes, the *K*_C_ values are represented as a diagram (Figure 6).

In terms of the values of the association constants, all the investigated complexes (Table 2) are considered extremely weak according to the classification reported by Carrier et al. [22]. Notably, the greater correlation coefficients and Fisher criteria were derived for the 4.0 pH buffer systems, indicating the higher quality of the linear correlations. As Table 2 and Figure 6 show, the complexes are more stable (approximately two-fold) in a buffer pH of 6.8 than those in a buffer pH of 4.0. This is an expected result because the uncharged species (even their small amount) appearing in the 6.8 pH medium more readily come in contact with the hydrophobic cyclodextrin cavity [23]. In addition, slightly greater K_C_ values were detected in the systems with SBE-β-CD than in those with HP-β-CD. Most likely, the structure and charge of the cyclodextrin molecule influenced the *K*_C_ regularities. On the one hand, the number of the hydrogen bond donors/acceptors was greater in the HP-β-CD molecule (25/39) than that in the SBE-β-CD one (21/35), which allowed us to suggest a stronger specific interaction of NTT•HCl with the former one. On the other hand, the charge–charge attraction and electrostatic interactions between the cationic species of the compound and anionic ones of SBE-β-CD [19] most likely tended to increase the complex stability constant. As a result, these two opposite trends are responsible for the *K*_C_ values, with the latter factor being determinative. The values of the stability of the complexes fully agree with the fact that the distribution coefficients tend to decrease gradually upon the CD concentration increasing (Figure 4). Obviously, as the NTT•HCl interaction with CD in the aqueous phase becomes stronger, the transition to the lipophilic medium (1-octanol) becomes more hampered.

### 2.2. Zeta Potential Evaluation

To approve the validity of the results, we evaluated the stability of the systems in a buffer pH of 6.8 in the presence of HP-β-CD and SBE-β-CD with the help of the zeta potential values measured using the light scattering experiments. The results are given as a diagram in Figure 7.

As follows from the diagram, according to the absolute zeta potential values, the presence of cyclodextrins facilitates the stability growth. Besides this, the negative signs of zeta potential, growing as the CD concentration increases, make it evident that the particles are more electrically stabilized at higher CD concentrations. As expected, this effect is more pronounced in the case of negatively charged SBE-β-CD, which agrees with the distribution results for $D_{app}^{oct/buf(pH6.8)}$, $D_{app}^{hex/buf(pH6.8)}$, ∆log*D* and association constants.

### 2.3. NTT•HCl Permeability, Effect of Cyclodextrin and Membrane Characteristics

In the present study, the permeability coefficients of NTT•HCl were calculated by measuring the rate of diffusion through two types of barriers. A regenerated MWCO 12–14 kDa (RC) cellulose membrane was applied as a model membrane. This membrane is often used to evaluate in vitro permeability when the aim is to determine the influence of additional components in the drug’s formulation on the permeability. The other membrane was the PermeaPad barrier (PP), which is applied to reveal the effect of some components of “real” biological membranes. This is possible because the PermeaPad barrier represents a bilayer structure of liposomes from soy phosphatidylcholine (S-100), which contains components that are characteristic of intestinal membranes, such as glycolipids, triglycerides, glycerol, fatty acids and choline. All permeation experiments were carried out at a pH of 6.8 and a pH of 7.4 for the donor and acceptor solutions, respectively, to simulate the compound’s transition through the intestinal membranes to the blood flow [24]. The permeability coefficients of NTT•HCl (*P_app_*), together with the raw experimental data represented as donor solution concentrations (*C*) and fluxes (*J*) at a steady penetration rate of the compound through the membrane, are listed in Appendix A. The diagram in Figure 8 demonstrates visually the similarities and differences between the permeability coefficients in a number of systems.

The data in Figure 8 suggest that cyclodextrins decrease the coefficients of NTT permeability through the investigated membranes. According to the literature [18], the penetration of this substance through the human epidermis in the presence of several chemical enhancers, such as polysorbate 80, ethanol, propylenglycol, ethanol and oleic acid, also decreased when a buffer pH of 7.4 was used, as opposed to a buffer pH of 5.5, for which the additives facilitated a penetration increase. The differences in the nature of the reported enhancers and cyclodextrins, as well as the nature of the membranes used in Melero et al.’s study [18] and in our work, make any explanations of the observed regularity rather speculative. However, assuming that the values of a pH of 6.8 and a pH of 7.4 are close, we can range the coefficients of NTT permeability through different membranes, as follows: epidermis (0.36 × 10^−6^) [18] << PermeaPad (2.44 × 10^−5^) < Cellulose membrane (3.90 × 10^−5^). This inequality clearly demonstrates a 67-fold greater resistance of the epidermis than that of the PermeaPad barrier.

Figure 8 demonstrates that NTT•HCl diffusion is faster in the case of the RC membrane, as expected, due to the presence of a lipophilic phospholipid layer in the PP barrier. A similar phenomenon was observed for amitryptiline hydrochloride (AMT•HCl), another tricyclic antidepressant and a structural analog of NTT•HCl [14]. Moreover, as Figure 8 shows, an essentially less-pronounced decrease in the coefficients of permeability through the RC membrane (than in those through the PP barrier) with CDs in the solution was observed. It is highly probable that the decrease in the rate of permeation through the RC membrane was caused by a slight reduction in the free compound molecules in accordance with the small values of the association constants of NTT with CDs: *K*_C_ (NTT•HP-β-CD) < *K*_C_ (NTT•SBE-β-CD) (Table 2, Figure 6). Undoubtedly, there are several factors responsible for the great extent of the permeability decrease in the case of the PermeaPad barrier and CDs in the solution (which is explained below). The effect of the lipophilic layer was found to be greater for AMT, as the differences between the coefficients of permeability through the PP and RC are 4.0-fold and 1.6-fold bigger for AMT and NTT, respectively. Evidently, this fact is in agreement with the higher distribution coefficients for AMT than those for NTT due to the presence of two hydrophobic substituents (CH_3_) in AMT instead of only one in the structure of NTT (see Section 2.1). As expected, this effect is more pronounced in the distribution process (as compared to permeation) because the range of distribution variations is usually much wider than that of permeability. The distribution coefficients of the drugs with similar permeability can differ from each other by 5–6 orders of magnitude [25].

### 2.4. Correlations of NTT•HCl PermeaPad Permeability and 1-Octanol/Buffer 6.8 pH Distribution

The processes of distribution in the 1-octanol/buffer 6.8 pH system and permeability through the PermeaPad barrier can help with understanding transport in biological membranes. Studying these processes in the presence of pharmaceutical excipients can be applicable to designing drug delivery systems. In the present work, to compare the distribution in the 1-octanol/buffer 6.8 pH system ($D_{app}^{oct/buf(pH6.8)}$) and the permeation through the PermeaPad barrier (*P*_app_ (PP)), we combined their dependencies on the CD concentration (Figure 9).

The regularities for $D_{app}^{oct/buf(pH6.8)}$ and *P_app_* (PP) in the presence of CDs are expected to be similar because the 1-octanol/water distribution coefficient serves as a descriptor, which, among others (such as size, ionization, hydrogen bonding capability, etc.), influences the permeability. As has been reported [25,26], the biological membrane, which is oil-like in nature, consists of amphiphilic molecules comprising the bilayer with the polar groups facing the exterior border of the aqueous phase and the lipid chains facing the bilayer center. As stated above, the lipophilic PermeaPad barrier contains components of biological membranes and forms a bilayer when it bulges, coming in contact with water and, thus, acquiring the properties of “real” membranes. In its turn, n-octanol containing a long alkyl chain and a functional group with hydrogen bond donor and acceptor properties is superficially similar to lipids [27]. The above-mentioned information explains the similar trends in the distribution and permeation processes both in the pure buffer and in the solution with CDs. Notably, a dramatic decrease in both the permeation rate and the extent of the distribution to the 1-octanol phase was observed when going from zero to the minimum CD concentration of 0.0115 M. There was no pronounced further decrease in either of the parameters. This situation is quite common and has been reported in the literature [6], including in studies by our research team [13,15,28]. The next point to be addressed is the stronger effect of SBE-β-CD (than that of HP-β-CD) on the permeability. As shown above for the distribution processes (Section 2.1), the effect of SBE-β-CD was more pronounced due to the electrostatic interactions between the NTT cationic species and anionic SBE-β-CD. On the one hand, the permeation is likely to be caused by the same factor, as the NTT•SBE-β-CD complexes are more stable than those with HP-β-CD, and neither cyclodextrin nor the complex with the drug can penetrate the PermePad barrier. On the other hand, when considering the permeability through the PermeaPad barrier, it is important to take into account the specific properties of the membrane. The structure of the membrane lipids suggests that there is a dipole layer between the aqueous phase and the hydrocarbon interior of the membrane, which leads to a negative surface potential [6]. This fact seems to suggest that the NTT transition to the surface of the membrane in the complex with negatively charged SBE-β-CD is hindered because of the repulsion of the charges. Thus, it is these two points that determine the maximal decrease in the permeability with SBE-β-CD in the solution.

## 3. Materials and Methods

### 3.1. Materials

Nortryptiline hydrochloride (NTT•HCl) (C_19_H_21_N•HCl) with a purity of ≥98% and sulfobutylether-β-CD with a purity of 99%, were purchased from BLDpharm (https://www.bld-pharm.com/ (accessed on 2 February 2023)). 2-Hydroxypropyl-β-cyclodextrin (purity ≥ 96%) was supplied by Sigma-Aldrich. 1-Octanol (purity ≥ 99%) and n-hexane (purity ≥ 0.97%) were obtained from Sigma-Aldrich. Potassium dihydrogen phosphate (purity ≥ 99%), disodium hydrogen phosphate dodecahydrate (purity ≥ 99%), sodium hydroxide (purity ≥ 98%), sodium acetate (purity ≥ 99%) and glacial acetic acid (purity ≥ 99%) were purchased from Merk (Darmstadt, Germany). All reagents and solvents were used as received.

The phosphate buffer with a pH of 7.4 was prepared in the following way: 23.6 g of Na_2_HPO_4_·12H_2_O was dissolved in H_2_O (1 L) (Solution 1), and 2.27 g of KH_2_PO_4_ was dissolved in 250 mL of H_2_O (Solution 2). Volumes of 808 mL (Solution 1) and 192 mL (Solution 2) were combined, and 167 mL of H_2_O was added to obtain an ionic strength of I = 0.15 mol·L^−1^. The phosphate buffer with a pH of 6.8 was made as follows: 27.22 g of KH_2_PO_4_ was dissolved in 1 L of water (Solution 1), and 2 g of NaOH was added to 250 mL of H_2_O (Solution 2). Amounts of 250 mL of Solution 1 and 112 mL of Solution 2 were mixed together and diluted with water to 1 L (I = 0.07 mol·L^−1^). The acetic buffer with a pH of 4.0 (I = 0.1 mol·L^−1^) was prepared using Solution A (822 mg of sodium acetate dissolved in 100 mL of water) and Solution B (1.44 mL of glacial acetic acid dissolved in 250 mL of water). Then, 100 ml of Solution B was titrated with 20 ml of Solution A. Bidistilled water (2.1 μS cm^−1^ electrical conductivity) was taken to prepare the buffer solutions. A FG2-Kit pH meter (Mettler Toledo, Im Langacher 44, 8606 Greifensee, Switzerland) standardized with 4.00 and 7.00 pH solutions was used to check the pH of the prepared buffers.

### 3.2. Distribution Experiments in Systems with and without Cyclodextrins

The apparent distribution coefficients of NTT•HCl ($D_{app}^{Org/buf}$) in the 1-octanol/buffer ($D_{app}^{oct/buf}$) and n-hexane/buffer ($D_{app}^{hex/buf}$) systems were measured at 37 °C via the standard shake-flask method [29] using mutually saturated organic and aqueous solvents, as reported in our earlier studies [14,15]. The experimental procedure was partially modified in the case of cyclodextrins (both HP-β-CD and SBE-β-CD) in the aqueous phase of the distribution system. Following the respective literature [17,30], in the present study, we prepared 1-octanol or n-hexane saturated with a buffer and a buffer saturated with 1-octanol and n-hexane. Three cyclodextrin concentrations (0.0115 M, 0.025 M and 0.035 M) in the aqueous phase were introduced. For the 1-octanol/buffer distribution, the examined solution of NTT•HCl was prepared in 1-octanol saturated with a buffer (C ~ 2 × 10^−3^ M), whereas for the n-hexane/buffer system, it was prepared in a buffer saturated with n-hexane (C ~ 2 × 10^−3^ M). Taking into account the obtained values of NTT•HCl solubility in the organic solvents and aqueous buffers, equal volumes of the 1-octanolic and buffer phases were prepared, whereas the n-hexane/buffer ratio was 12:2. Two pH values of the aqueous phases were examined: a pH of 4.0 and a pH of 6.8, simulating different segments of the gastro-intestinal tract. The respective volumes of the phases were placed into glass vials and equilibrated in an air thermostat for 3 days at 37 °C. Then, the phases were separated after at least 6 h of storage. The concentrations in both phases were measured using a UV–Vis spectrophotometer (Cary-50, USA Cary-50 Varian, Palo Alto, CA, USA, Software Version 3.00 (339)) with the help of the calibration curves. Each distribution test was carried out at least three times. The resultant values of the distribution coefficients were from an average of at least four replicated experiments with an accuracy of 2–4%.

The distribution coefficients ($D_{app}^{Org/buf}$) were determined, taking into account the compound concentrations in the organic ($C_{2}^{Org}$) and buffer ($C_{2}^{buf}$) phases and the volumes of the buffer ($V_{}^{buf}$) and organic ($V_{}^{Org}$) phases:(1)$$D_{app}^{Org/buf}=\frac{C_{2}^{Org}\cdot V^{buf}}{C_{2}^{buf}\cdot V^{Org}}$$

The ∆log*D* parameter of Seiler [11] was calculated for a single CD concentration in the aqueous phase, equal to 0.0115 M for both HP-β-CD and SBE-β-CD, as follows:(2)$$\Delta \log D=\log D_{app}^{oct/buf}-\log D_{app}^{hex/buf}$$

Based on the distribution results, we determined the association constants of NTT•HCl with both cyclodextrins. To this end, the phase distribution approach fully described in [17] was applied. An assumption was made that both cyclodextrins used did not partition to the organic phases due to their high aqueous solubility, reported by Saokham et al. [31] to be >1200 mg·mL^−1^. It is evident that, in the presence of cyclodextrins, the distribution coefficient of the compound can be described by the following equation:(3)$$D_{app}^{Org/(buf+CD)}=\frac{C_{2}^{Org}(drug)}{C_{2}^{buf}(drug)+C_{2}^{buf}(drug•CD)}$$
where $D_{app}^{Org/(buf+CD)}$ is the distribution coefficient of the drug in the presence of CD, $C_{2}^{Org}$ and $C_{2}^{buf}$ are the drug concentrations in the organic and aqueous phases, respectively, and $C_{2}^{buf}(drug•CD)$ is the concentration of the drug/cyclodextrin complex in the aqueous phase of the distribution system. As follows from Equation (3), the association constant of the complex can be derived from $D_{app}^{Org/buf}$ and $D_{app}^{Org/(buf+CD)}$ at specific CD concentrations using Equation (4).
(4)$$\frac{D_{app}^{Org/(buf+CD)}}{D_{app}^{Org/buf}}=1+K_{C}\cdot C_{CD}$$

It can be performed by plotting the following dependence:(5)$$\log(\frac{D_{app}^{Org/buf}-D_{app}^{Org/(buf+CD)}}{D_{app}^{Org/buf}})=\log(K_{C})+\alpha \cdot \log(C_{CD})$$

### 3.3. Determination of Permeability in Systems with and without Cyclodextrins

The permeability coefficients of NTT•HCl were determined in the absence and in the presence of cyclodextrins (HP-β-CD and SBE-β-CD) in the donor solution. A model artificial regenerated cellulose membrane MWCO 12–14 kDa (Standard Grade RC Dialysis Membrane, Flat Width 45 mm) (RC) or a biomimetic PermeaPad barrier (PHABIOC, Germany, www.permeapad.com (accessed on 1 February 2023)) (PP) were placed between the donor and receptor compartments of a vertical type Franz diffusion cell (PermeGear, Inc., Hellertown, PA, USA) with 7 mL/1 mL volumes for the donor and acceptor solutions. The cyclodextrin concentrations were 0.0115 M, 0.025 M and 0.035 M when the RC membrane was used and only 0.035 M in case of the PP barrier. Two different pH values (pH of 4.0 and pH of 6.8) were applied in the donor solution, whereas the receptor chamber was filled with a buffer with a pH of 7.4, modeling the compound’s diffusion to the blood plasma in all the experiments. The effective surface area of the membrane was 0.785 cm^2^. The temperature of 37 °C (as in the distribution experiments) was maintained. Samples with a volume of 0.5 mL for the receptor solution were withdrawn every 30 min for 5 h and analyzed with a spectrophotometer (Spectramax 190; Molecular Devices Corporation, San Jose, CA, USA) in 96-well UV black plates (Costar, Washington, DC, USA) at a wavelength of 239 nm. The fluxes in the steady state (*J*) were determined from the kinetic dependencies of the cumulative amount of the permeated drug (*Q*) taking into account the effective surface area of the barrier (*A*) via the following equation:(6)$$J=\frac{dQ}{A\cdot dt}$$

The permeability coefficient was calculated using the NTT•HCl concentration in the donor solution (*C*_0_), as follows:(7)$$P_{app}=\frac{J}{C_{0}}$$

The values of the experimental permeability coefficients corresponded to an average of at least three replicas with an accuracy up to 4%. The sink conditions were realized throughout each experiment. This means that the drug concentration in the acceptor chamber did not exceed 10% of that in the donor chamber at any time.

### 3.4. Light Scattering Examination

Zetasizer Nano-ZS (Zetasizer Nano ZS, Malvern Instruments, Worcestershire, UK) was employed to perform the light scattering measurements at a scattering angle of 90°. A He–Ne gas laser operating at 633 nm was used as the light source. The donor solutions of NTT•HCl examined in the permeability experiments were subjected to DLS. Each experiment was repeated at least 3 times. In addition, the zeta potential was determined using the Smoluchowski approximation.

## 4. Conclusions

In the present study, the effect of two cyclodextrins on the distribution and permeation processes of nortryptiline hydrochloride (NTT•HCl), an antidepressant drug of the tricyclic class, was disclosed. The distribution experiments were carried out using two model systems of immiscible solvents: 1-octanol/buffer (pH of 7.4 and pH of 4.0) and n-hexane/buffer (pH of 7.4 and pH of 4.0) in the absence and in the presence of 2-hydroxypropyl-β-cyclodextrin (HP-β-CD) and sulfobutylether-β-cyclodextrin (SBE-β-CD) of 0.0115 M, 0.025 M and 0.035 M concentrations in the aqueous phases at 37 °C. The equilibrium was shifted to the aqueous phase in all studied n-hexane/buffer systems, regardless of the pH and presence of CDs. During the distribution examination, we evaluated the reduction in the NTT•HCl transfer from the aqueous to both organic phases in the presence of cyclodextrins. The effect of SBE-β-CD was shown to be more pronounced than that of HP-β-CD. The Δlog*D* parameter proved the weaker effect of hydrogen bonding on the transfer process in the cyclodextrin-containing systems. The slightly greater values of the association constants (*K*_C_) in SBE-β-CD than those in HP-β-CD fully agree with the much lower values of the distribution coefficients in the presence of SBE-β-CD, determined via the phase distribution method. The zeta potential measurements made by the light scattering method revealed elevated stability of the system upon CD concentration growth, especially in SBE-β-CD.

The permeation experiments carried out on the regenerated cellulose membrane (RC) and the lipophilic PermeaPad barrier (PP) composed of soy lecithin demonstrated faster diffusion through the RC membrane. Moreover, an essentially less-pronounced decrease in the permeability coefficients was revealed with CDs in the solution in case of the RC membrane than that when the PP barrier was applied. This effect was again stronger in SBE-β-CD, which was attributed to the electrostatic interactions between the cationic NTT species and anionic SBE-β-CD and the negative surface potential of the PP barrier. Similar trends in the variations in the 1-octanol/buffer 6.8 pH distribution coefficients ($D_{app}^{oct/buf(pH6.8)}$) and coefficients of permeability through the PermeaPad barrier (*P_app_* (PP)) in the presence of CDs were disclosed and explained by the similarity (to some extent) between the structures of the amphiphilic molecules comprising the bilayer of the PP barrier and n-octanol containing a long alkyl chain and a functional group with hydrogen bond donor and acceptor properties.

Judging according to the obtained results, it can be concluded that both the distribution and permeability approaches are helpful for designing cyclodextrin-based drug formulations. Moreover, using the lipophilic PermeaPad barrier is advantageous, as it makes it possible to account for the interactions of both the drug and the excipients, on the one hand, and the components of “real” biological membranes, on the other. We hope that the presented results improve the understanding of drug delivery using cyclodextrin-containing systems.

## 5. Future Prospects and Limitations

In our future studies, we would like to disclose the causes of the permeability variations in the presence of a variety of solubilizing agents, including co-solvents, polymeric micelles, cyclodextrins and their combinations. To this end, we propose adapting the mechanistic approach described in [32] to gain deeper insight into the diffusion/permeation of other tricyclic antidepressant drugs by evaluating a truly molecularly dissolved drug fraction. In our opinion, overlooking this issue would make it more difficult to interpret the obtained results.

Another issue to be addressed is the simultaneous investigation of the dissolution and permeation of a drug in the presence of solubilizing agents in non-sink conditions using the D/P setup in a side-by-side diffusion cell. As a result, we intend to obtain information about the variations in the drug’s flux through the membrane over time.

## Figures and Tables

**Figure 1 pharmaceuticals-16-01022-f001:**
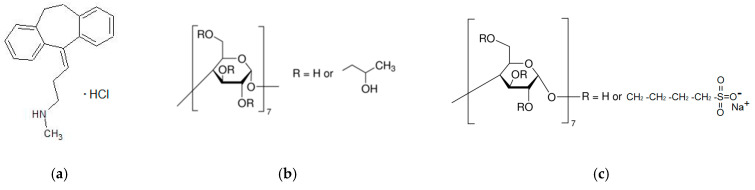
Nortryptiline hydrochloride (NTT•HCl) (**a**), 2-hydroxypropyl-β-cyclodextrin (HP-β-CD) (**b**) and sulfobutylether-β-cyclodextrin (SBE-β-CD) (**c**) structures.

**Figure 2 pharmaceuticals-16-01022-f002:**
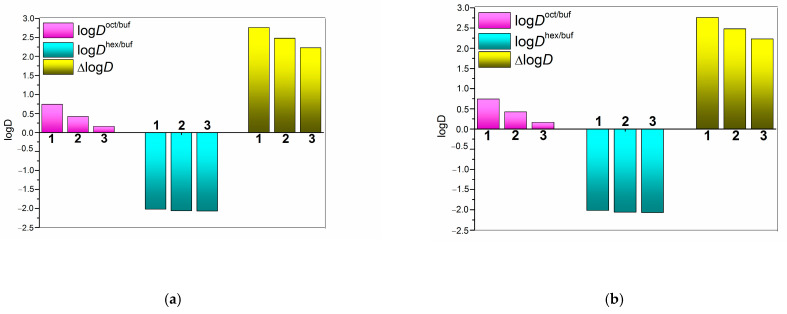
Distribution coefficients, log$D_{app}^{oct/buf}$, log$D_{app}^{hex/buf}$, and ΔlogD parameter without cyclodextrins (1), with 0.0115 M of HP-β-CD (2) and with 0.0115 M of SBE-β-CD (3) in the aqueous phase for NTT•HCl at 37 °C: (**a**) pH of 6.8, (**b**) pH of 4.0 of the buffer phase.

**Figure 3 pharmaceuticals-16-01022-f003:**
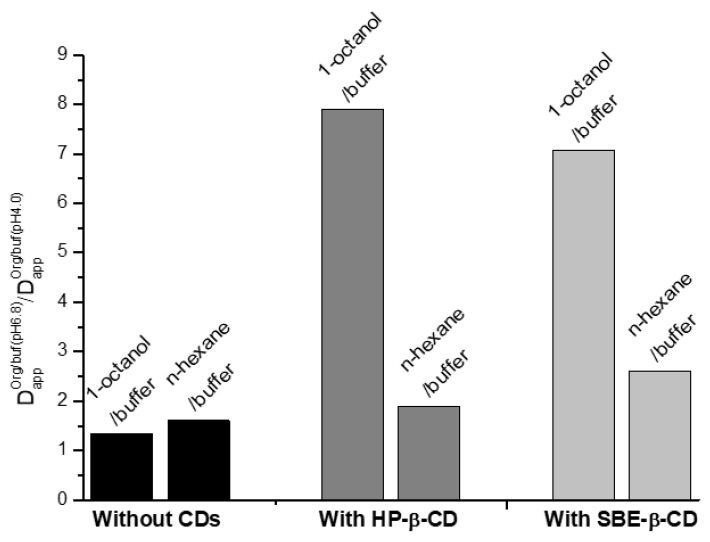
Diagram illustration of the ratios between the distribution coefficients at pH of 6.8 ($D_{app}^{Org/buf(pH6.8)}$) and pH of 4.0 ($D_{app}^{Org/buf(pH4.0)}$) of the buffer phase; C_CD_ = 0.0115 M.

**Figure 4 pharmaceuticals-16-01022-f004:**
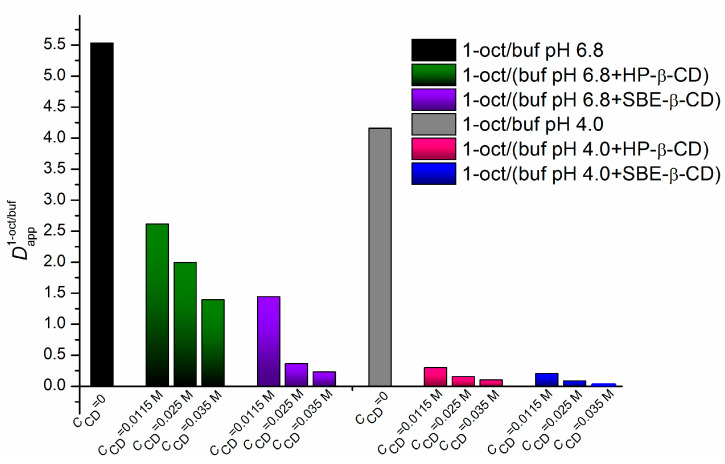
Diagram illustrating the trends of the variations in the distribution coefficients in the 1-octanol/buffer system ($D_{app}^{oct/buf}$) following the CD concentration growth at different pH values of the aqueous phases; the cyclodextrin concentrations are shown under the columns; 37 °C.

**Figure 5 pharmaceuticals-16-01022-f005:**
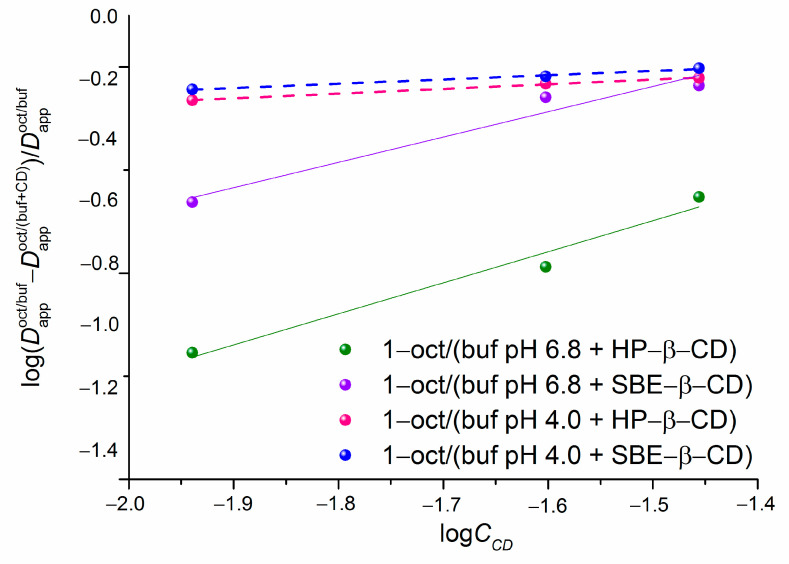
Plots of the dependencies used to calculate the association constants of NTT•HCl with cyclodextrins at a pH of 6.8 and a pH of 4.0 versus the experimental distribution coefficients in the 1-octanol/buffer system, log scale, 37 °C; the straight lines refer to a buffer pH of 6.8; the dash lines refer to a buffer pH of 4.0.

**Figure 6 pharmaceuticals-16-01022-f006:**
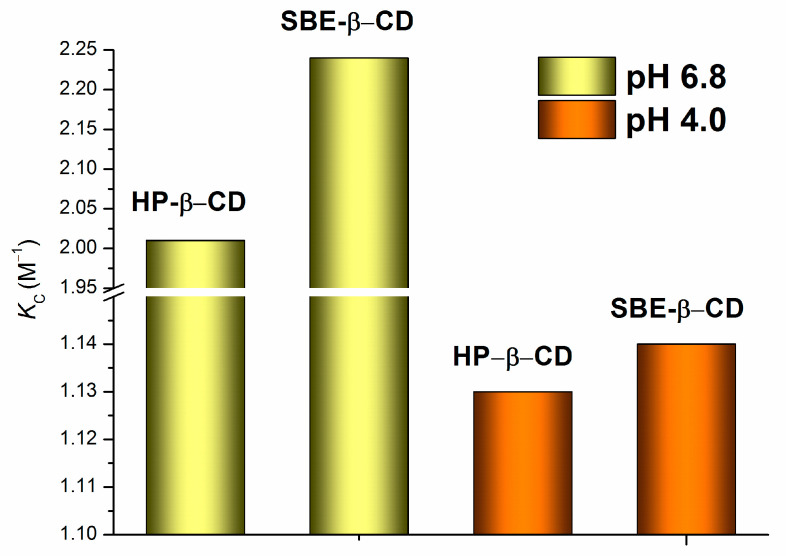
Comparison of the association constants of NTT•HCl with HP-β-CD and SBE-β-CD at a pH of 4.0 and a pH of 6.8 of the aqueous phase of the 1-octanol/buffer system, 37 °C.

**Figure 7 pharmaceuticals-16-01022-f007:**
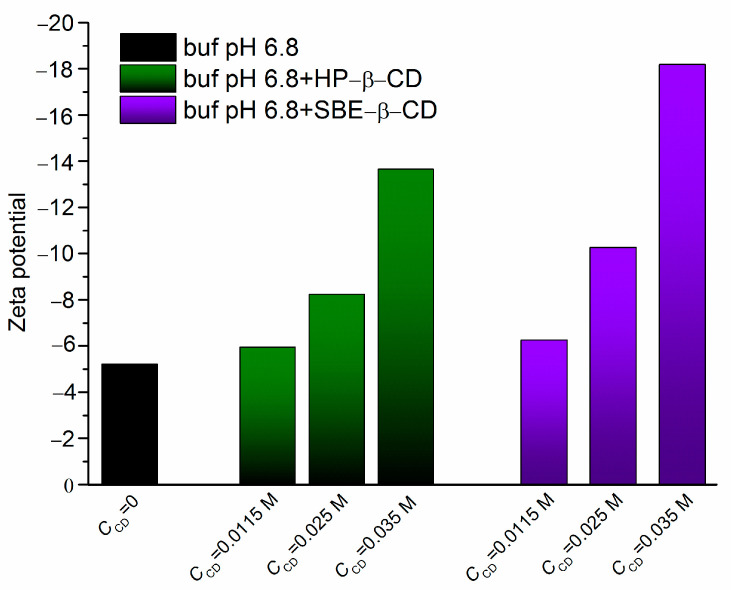
Zeta potentials of NTT•HCl solutions in the presence of HP-β-CD and SBE-β-CD in comparison with those in pure buffer pH of 6.8. The CD concentrations are indicated under the columns of the diagram.

**Figure 8 pharmaceuticals-16-01022-f008:**
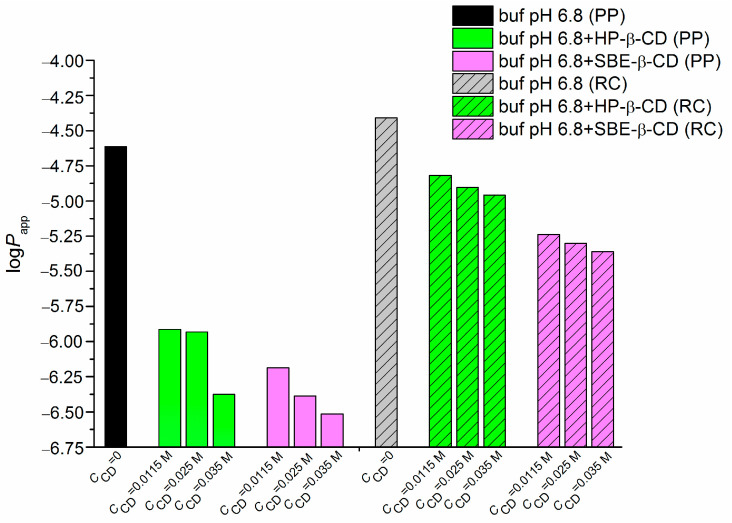
Coefficients of NTT•HCl permeability through the PermeaPad barrier (PP) and regenerated cellulose membrane (RC) in the absence and in the presence of HP-β-CD and SBE-β-CD; logarithmic scale; 37 °C.

**Figure 9 pharmaceuticals-16-01022-f009:**
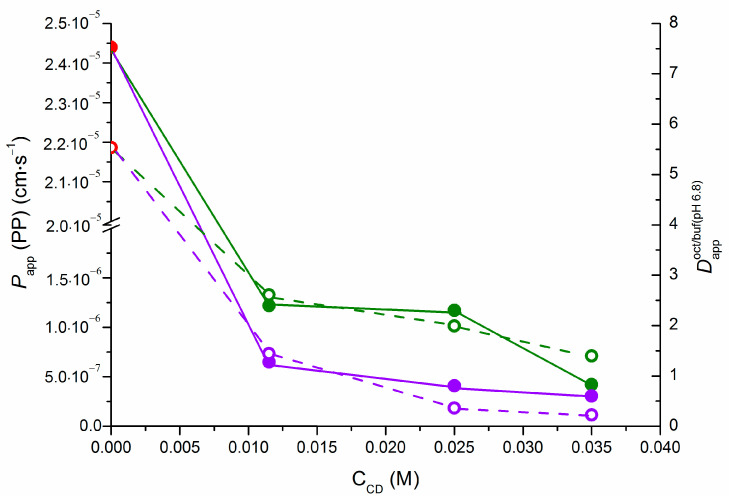
Coefficients of permeability through the PermeaPad barrier (*P_app_* (PP)) and distribution coefficients ($D_{app}^{oct/buf(pH6.8)}$) at different CD concentrations: *P_app_* (PP) is denoted by filled points and straight lines, $D_{app}^{oct/buf(pH6.8)}$ is shown by empty points and dashed lines; HP-β-CD is represented in green, SBE-β-CD is represented in violet.

**Table 1 pharmaceuticals-16-01022-t001:** Distribution coefficients, $D_{app}^{Org/buf}$ (log$D_{app}^{Org/buf}$ ), of NTT•HCl in the 1-octanol/buffer and n-hexane/buffer systems at pH of 6.8 and pH of 4.0 without and with HP-β-CD and SBE-β-CD in the aqueous phase, and ∆log*D* parameter, 37 °C.

*C*_CD_ (M)	$D_{app}^{oct/buf}$ $\;(\log D_{app}^{oct/buf})$	$D_{app}^{hex/buf}$ $\;(\log D_{app}^{hex/buf})$	∆log*D*
	pH of 6.8		
0	5.537 ± 0.111 (0.74)	(9.5 ± 0.2) × 10^−3^ (−2.02)	2.76
	HP-β-CD		
0.0115	2.613 ± 0.022 (0.42)	(8.7 ± 0.1) × 10^−3^ (−2.06)	2.48
0.025	1.993 ± 0.030 (0.30)	-	
0.035	1.395 ± 0.025 (0.15)	-	
	SBE-β-CD		
0.0115	1.443 ± 0.040 (0.16)	(8.6 ± 0.3) × 10^−3^ (−2.07)	2.23
0.025	0.364 ± 0.011 (−0.44)	-	
0.035	0.227 ± 0.006 (−0.64)	-	
	pH of 4.0		
0	4.159 ± 0.080 (0.62)	(5.6 ± 0.2) × 10^−3^ (−2.23)	2.85
	HP-β-CD		
0.0115	0.299 ± 0.008 (−0.52)	(4.6 ± 0.1) × 10^−3^ (−2.34)	1.82
0.025	0.154 ± 0.003 (−0.81)	-	
0.035	0.101 ± 0.001 (−1.0)	-	
	SBE-β-CD		
0.0115	0.204 ± 0.005 (−0.69)	(3.3 ± 0.1) × 10^−3^ (−2.48)	1.79
0.025	0.086 ± 0.001 (−1.07)	-	
0.035	0.013 ± 0.000 (−1.89)	-	

Standard uncertainties *u*(*t*) = 0.2 °C, *u*(pH) = 0.02 pH units; the distribution coefficient values represent the mean ± SD (n ≥ 4); the relative standard uncertainties *u*_r_(*D_app_*) = 0.04 were determined from the standard uncertainties *u*(*D_app_*) divided by the mean value of the distribution coefficient: *u*_r_(*D_app_*) = *u*(*D_app_*)/│*D_app_*│.

**Table 2 pharmaceuticals-16-01022-t002:** Association constants of the NTT•HCl/cyclodextrin complexes derived via the phase distribution method in 1-octanol/buffer systems at 37 °C and regression parameters of the linear dependencies.

pH of the Aqueous Phase	*K*_C_ (M^−1^)	^1^ *R*	^2^ σ	^3^ *F*
	HP-β-CD			
pH of 6.8	2.01 ± 0.05	0.9866	3.06 × 10^−4^	36.4
pH of 4.0	1.13 ± 0.02	0.9991	4.69 × 10^−7^	538.9
	**SBE-β-CD**			
pH of 6.8	2.24 ± 0.03	0.9784	3.30 × 10^−4^	22.4
pH of 4.0	1.14 ± 0.02	0.9965	1.49 × 10^−6^	141.6

^1^ pair correlation coefficient; ^2^ residual sum of squares; ^3^ Fisher criterion.

## Data Availability

No data were used for the research described in this article.

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
