# Peer review of "Cyclodextrin’s Effect on Permeability and Partition of Nortriptyline Hydrochloride"

_pharmaceuticals, 2023, doi:10.3390/ph16071022_

Round 1

Reviewer 1 Report

The manuscript proposes a very interesting study on cyclodextrins for membrane permeability and partition processes of nortriptyline hydrochloride aimed to drug delivery optimization.
The topic correlates to the journal.
The work could be better structured, although the core idea is novel and interesting: for instance the title requires to be re-structured and shortened for a better and quicker understanding.
Therefore, there are some minor issues that require to be addressed before proceeding with the publication, to enhance the quality and presentation to a broad audience.
The abstract reports a consistent summary of the article core, but it spans in a very scattered way so that the reader might be in trouble by trying to understand it. It is suggested to schematically proceed with i.e. a sentence recapitulating the cyclodextrines family, drug delivery potentials so far, the core work/aim and the achieved results.
Please, rephrase the introduction as well: it reports on the same criticism of the abstract. That is very confusing and might decrease the article visibility.
Moreover, aim and hypothesis are missing: it is suggested to structure them at the very end of the introduction, in order to create a straightforward flow throughout the article, for the readers’ benefit.
No statistics is provided: based on the amount of replicas it is assumed the Authors provided for it, therefore it is largely suggested for them to add on the related details (test, software, etc).
Please, better describe the details of the comparison with other buffers (I.e. dmem, physiological buffer/PBS)  for the system to exert its own purpose: no satisfactory discussion and related references are provided, which are mandatory in order to define the scientific context. At this purpose, references list must be improved with additional, broader state-of-the-art sources, in order to enhance the core discussion (e.g. doi: 10.1016/j.cis.2017.05.009; 10.1016/j.ijpharm.2018.07.026), mainly in the introduction section and decrease the self-citation rate.
A ToC graphic would definitely help.
References format must be consistent and all journal must be correctly abbreviated, as well as fonts to be used: please, amend (e.g., ref. #14).
An English check would strongly boost the whole manuscript (e.g., replace “well known”, “At that”, “Interrelation”, “Many parent cyclodextrins”, and shortened long sentences such as “Assessment of drug permeation to the brain may be crucial in several cases, for example, for the drugs exhibiting an undesirable hypnotic effect which should be minimized; or for the drugs aimed at the treatment of neurodegenerative and mental disorders which, as opposed to the previous case, should be timely delivered to the brain (to pass through not only the intestinal membranes but also the blood-brain barrier).”)
Check for typos.
Moreover, it is suggested to list the abbreviations as a point-listed: it will boost the scientific appeal.  
It might help discuss over an additional section any
possible limitations/future perspectives.

An English check would strongly boost the whole manuscript (e.g., replace “well known”, “At that”, “Interrelation”, “Many parent cyclodextrins”, and shortened long sentences such as “Assessment of drug permeation to the brain may be crucial in several cases, for example, for the drugs exhibiting an undesirable hypnotic effect which should be minimized; or for the drugs aimed at the treatment of neurodegenerative and mental disorders which, as opposed to the previous case, should be timely delivered to the brain (to pass through not only the intestinal membranes but also the blood-brain barrier).”) An English native speaker check is mandatory.

Author Response

Reply to Reviewer 1 comments:

Dear Reviewer, thank you very much for the careful consideration of our manuscript. We've taken into account all your comments and suggestions. The corrections are marked in red in the manuscript body.

Comment:

The work could be better structured, although the core idea is novel and interesting: for instance the title requires to be re-structured and shortened for a better and quicker understanding.
Therefore, there are some minor issues that require to be addressed before proceeding with the publication, to enhance the quality and presentation to a broad audience.

Reply:

The title has been re-structured and shortened for a better and quicker understanding.

Comment:

The abstract reports a consistent summary of the article core, but it spans in a very scattered way so that the reader might be in trouble by trying to understand it. It is suggested to schematically proceed with i.e. a sentence recapitulating the cyclodextrines family, drug delivery potentials so far, the core work/aim and the achieved results.

Reply:

The abstract has been re-written according to the Reviewer’s recommendations.

Comment:

Please, rephrase the introduction as well: it reports on the same criticism of the abstract. That is very confusing and might decrease the article visibility.

Reply:

The introduction has been re-written according to the Reviewer’s recommendations.

Comment:

Moreover, aim and hypothesis are missing: it is suggested to structure them at the very end of the introduction, in order to create a straightforward flow throughout the article, for the readers’ benefit.
Reply:

The aim and hypothesis have been structured.

Comment:

No statistics is provided: based on the amount of replicas it is assumed the Authors provided for it, therefore it is largely suggested for them to add on the related details (test, software, etc).

Reply:

The missed details on the statistics, amount of replicas and accuracy for the distribution and permeability experiments have been provided according to the Reviewer’s recommendations: in "Materials and Methods" and in the footnotes to the tables in "Results and Discussion".

Comment:

Please, better describe the details of the comparison with other buffers (I.e. dmem, physiological buffer/PBS)  for the system to exert its own purpose: no satisfactory discussion and related references are provided, which are mandatory in order to define the scientific context. At this purpose, references list must be improved with additional, broader state-of-the-art sources, in order to enhance the core discussion (e.g. doi: 10.1016/j.cis.2017.05.009; 10.1016/j.ijpharm.2018.07.026), mainly in the introduction section and decrease the self-citation rate.

Reply:

A discussion of the comparison with the literature data and other buffers has been provided in the "Introduction" and "Results and Discussion". The reference list has been improved with additional sources in order to enhance the discussion according to the Reviewer’s recommendations.

Comment:

A ToC graphic would definitely help.
Reply:

A ToC graphic has been provided.

Comment:

References format must be consistent and all journal must be correctly abbreviated, as well as fonts to be used: please, amend (e.g., ref. #14).
Reply:

The references format and journal abbreviations have been corrected.

Comment:

An English check would strongly boost the whole manuscript (e.g., replace “well known”, “At that”, “Interrelation”, “Many parent cyclodextrins”, and shortened long sentences such as “Assessment of drug permeation to the brain may be crucial in several cases, for example, for the drugs exhibiting an undesirable hypnotic effect which should be minimized; or for the drugs aimed at the treatment of neurodegenerative and mental disorders which, as opposed to the previous case, should be timely delivered to the brain (to pass through not only the intestinal membranes but also the blood-brain barrier).”)
Reply:

The language of the manuscript has been thoroughly checked and edited.

Comment:

Check for typos.
Reply:

The manuscript has been checked for typos.

Comment:

Moreover, it is suggested to list the abbreviations as a point-listed: it will boost the scientific appeal.  
Reply:

The list of the abbreviations has been inserted in the manuscript at the end of the Introduction.

Comment:

It might help discuss over an additional section any
possible limitations/future perspectives.

Reply:

An additional section devoted to future prospects has been added immediately after the "Conclusions"

Reviewer 2 Report

Dae authors

Work have intersting point of view ans it is very interesting read.

Comments:

Introduction:

Line 74-75 sentences are not clear. Pleas rewrite.

Pleas add influence of pH to excipient ibtereaction and permeability in introduction

Results and discussion

Why buffer pH 4 and 6.8 were choosen

Correlation between octanol/buffer and mebranes should be made comparing similarity and differences between  results and applicabilility in formulation developement.

Conclusion should be made what method or more should be applied during formulation developement.

Also how it could be transfered to other API. Should they be similar structer. If not what should be performed so that permeability with solubility could be regulary applied in drug formulation. When you propose this approache to be applayed 

Materials

How you choose polarity of buffer?

Have you accounted ionic strength?

Conclusion

Membarne and octanol/buffer permeability shpuld be discussed jointly

Best of luck

Author Response

Reply to Reviewer 2 comments:

Dear Reviewer, thank you very much for careful consideration of our manuscript. We've taken into account all your comments and suggestions. The corrections are marked in red in the manuscript body.

Introduction:

Comment:

Line 74-75 sentences are not clear. Pleas rewrite.

Reply:

The sentences have been rewritten.

Comment:

Pleas add influence of pH to excipient ibtereaction and permeability in introduction

Reply:

The required information about the influence of pH on the interaction of the drug with the excipient and permeability has been added to the "Introduction" according to the Reviewer’s recommendations.

Results and discussion

Comment:

Why buffer pH 4 and 6.8 were choosen

Reply:

There were two reasons for the choice of these buffers for the experiments. On the one hand, these pHs are characteristic of the intestinal fluids where the main absorption of the drug takes place. On the other, differences in the NTT ionization in these media allowed us to reveal the effect of the ionization state on the investigated processes.

This explanation has been introduced into the manuscript (Results and Discussion).

Comment:

Correlation between octanol/buffer and mebranes should be made comparing similarity and differences between results and applicabilility in formulation developement.

Reply:

To underline the importance of the comparison between the 1-octanol/buffer distribution and permeability through the PermeaPad barrier we have organized this issue into a separate Section: 

2.4. Correlations of NTT•HCl PermeaPad permeability and 1-octanol/buffer pH 6.8 distribution.

The importance of this issue for the formulation development has been emphasized.

Comment:

Conclusion should be made what method or more should be applied during formulation developement.

Reply:

An additional conclusion has been added to the "Conclusions" section.

Comment:

Also how it could be transfered to other API. Should they be similar structer. If not what should be performed so that permeability with solubility could be regulary applied in drug formulation. When you propose this approache to be applayed 

Reply:

We believe the proposed approach can be applied to structurally similar API. Meanwhile, for its regular application we assume that it is necessary to find the permeability variations in the presence of different solubilizing agents using the mechanistic approach through the evaluation of a truly molecularly dissolved drug fraction.

To clarify this issue and shed light on our future plans we have introduced an additional section (6. Future prospects and limitations).

Materials

Comment:

How you choose polarity of buffer?

Have you accounted ionic strength?

Reply:

The polarity of the buffers was accounted for through the ionic strength calculation.

The ionic strength of the buffer solutions was calculated taking into account the concentrations of all the species of the buffer components. For clarity, a more detailed description of the preparation procedure for buffers pH 7.4 and pH 6.8 has been provided in the "Materials and methods" Section. The values of the ionic strength have also been introduced.

Conclusion

Comment:

Membarne and octanol/buffer permeability shpuld be discussed jointly

Reply:

A paragraph concerning the comparison of distribution and permeability has been introduced at the end of the "Conclusions".

Round 2

Reviewer 1 Report

The manuscript looks remarkably improved, therefore it is now suitable for publication.